# Advances in Imaging Techniques for Mammalian/Human Ciliated Cell’s Cilia: Insights into Structure, Function, and Dynamics

**DOI:** 10.3390/biology14050521

**Published:** 2025-05-08

**Authors:** Jin Li, Shiqin Huang, Hao Chen

**Affiliations:** 1Institute of Reproductive Medicine, Medical School of Nantong University, Nantong 226001, China; jing040899@163.com (J.L.);; 2The Guangdong-Hong Kong-Macao Joint Laboratory for Cell Fate Regulation and Diseases, GMU-GIBH Joint School of Life Sciences, Guangzhou Medical University, Guangzhou 511436, China; 3Key Laboratory of Reproductive Health Diseases Research and Translation (Hainan Medical University), Ministry of Education, Hainan Provincial Key Laboratory for Human Reproductive Medicine and Genetic Research, Hainan Provincial Clinical Research Center for Thalassemia, The First Affiliated Hospital of Hainan Medical University, Hainan Medical University, Haikou 570105, China

**Keywords:** cilia, imaging techniques, volume electron microscopy

## Abstract

This review summarizes recent advancements in imaging techniques used for the study of cilia, specialized hair-like structures projecting from the cell surfaces. Cilia play crucial roles in various biological processes, including sensory perception and cellular motility. This review highlights how innovative imaging methods, such as super-resolution microscopy and volume electron microscopy, have significantly advanced our understanding of cilia’s structure and function. These cutting-edge techniques have revealed detailed insights into the complexity of cilia architecture, underscoring their important roles in health and disease. By bridging basic research and clinical applications, this work enhances our knowledge of cilia-related disorders and provides valuable guidance for future functional investigations, potentially informing novel therapeutic strategies for cilia-related diseases.

## 1. Introduction

Cilia are microtubule-based, hair-like structures extending from the apical surface of eukaryotic cells, where they play critical roles in embryonic development, tissue homeostasis, and organismal physiology. These organelles mediate essential biological processes, including mucociliary clearance in the respiratory tract [1,2,3], gamete transport [4], the establishment of left–right body axis patterning [5,6], and cerebrospinal fluid circulation [7]. Although these structures were first observed in the 1770s [8], their ultrastructural complexity and molecular composition remain incompletely resolved.

This review aims to provide a comprehensive overview of recent advances in our understanding of ciliary biology, focusing on structural and functional insights from cutting-edge imaging technologies. By critically evaluating and comparing the merits and limitations of different imaging techniques in resolving ciliary architecture, dynamics, and dysfunction, we aim to offer a more integrated perspective on the role of cilia in both physiological and pathological contexts.

## 2. Fundamental Properties of Cilia

### 2.1. Definition, Structure, and Function

Cilia are microtubule-based organelles that protrude from the cell surface in a ‘hair-like’ structure and are found in most eukaryotes cell types, though their presence varies depending on cellular specialization. They serve as signaling hubs, housing receptors and functioning as motile organelles essential for cellular and organismal processes [9]. Typically, cilia range in length from a few to a dozen micrometers and in diameter from 0.2 to 0.5 micrometers. Their length is precisely regulated, reflecting the longitudinal arrangement of their microtubules. Cilia are classified into two categories based on their motility: primary cilia and motile cilia (also referred to as flagella) [10,11,12].

The basic composition of the cilia is remarkably evolutionarily conserved among species, and its core component consists of several highly conserved units: the ciliary membrane, axoneme, transition zone (TZ), and basal body (BB) [13]. The BB, derived from the centriole, anchors the cilia to the apical cytoplasm and plays a pivotal role in the spatiotemporal regulation of the cilia assembly process. The microtubule-based axoneme extends outward from the cell surface, forming the core structure of the cilium. The ciliary membrane is derived from the cell membrane, with the two membranes structurally interconnected. The axoneme exhibits two main structural patterns in the cross-section. In motile cilia, the axoneme follows a ‘9 + 2’ arrangement, consisting of nine outer microtubule doublets surrounding a central pair of microtubules. In contrast, primary cilia follow a ‘9 + 0’ arrangement, lacking the central microtubules. Transition fibers, located at the base of the cilium, connect the ciliary membrane to the basal body. The TZ, composed of protein complexes and Y-links, plays a crucial role in maintaining ciliary structure and function. Additionally, the intraflagellar transport (IFT) system, comprising IFT-A and IFT-B protein complexes, mediates the bidirectional transport of molecules along the cilium via kinesin-2 (anterograde) or dynein-2 (retrograde) motors [13,14] (Figure 1).

Motile cilia exhibit oscillatory movements, either rhythmic, wave-like, or planar, depending on their ‘9 + 2’ microtubule configuration, which enables them to agitate surrounding fluids and induce directional flow [15,16]. This structural design underpins the motile functions of cilia, including locomotion, facilitating cellular movement, and modifying the fluid environment through oscillation [17,18]. For instance, in the efferent ducts (EDs) of the male reproductive tract, cilia drive the directional flow of luminal contents through coordinated rhythmic fluctuations. This pattern of movement not only enhances the efficiency of sperm migration along the reproductive tract, but also reduces cell aggregation through mechanical perturbation, thereby significantly reducing the risk of lumen occlusion due to sperm retention [19]. Similarly, the sperm flagellum, a specialized motile cilia, enables sperm motility through its oscillatory motion, serving as a key indicator of sperm motility. Primary cilia, by contrast, are primarily involved in the detection of extracellular stimuli and the regulation of downstream signaling pathways. These components mediate complex signaling pathways, including GPCR and Ca^2+^ signaling, enabling cells to detect and respond to diverse stimuli. Primary cilia play tissue-specific sensory roles, including mechanosensors in vascular endothelial cells and bones, chemosensors in the nasal cavity, and sensing electrical signals in the nervous system [17,18,20]. Interestingly, studies suggest that motile cilia may also have signal-sensing capabilities, challenging their traditional classification as motile structures [21]. For example, in the mammalian oviduct epithelium, motile cilia are thought to respond to external stimuli such as sex hormones and mechanical signals, potentially contributing to the regulation of the internal environment [22].

### 2.2. Assembly Dynamics

Cilia are dynamic organelles that undergo structural and functional changes in response to the cell cycle. Their formation begins during the G_0_/G_1_ phase of the cell cycle [23]. Primary ciliogenesis is highly programmed with spatiotemporal characteristics and involves several key steps: centriole/BB anchoring, dissociation of centriolar cap proteins, ciliary bud formation, and the extension of axoneme facilitated by IFT complexes. In some cells, centrioles migrate directly to the plasma membrane, where they anchor and convert to BB via the extracellular pathway. In others, ciliogenesis occurs through an intracellular pathway that relies on the formation of a specialized structure—the ciliary pocket—formed by membrane invagination. The intracellular pathway involves initiation, ciliary bud formation, and axoneme extension.

During initiation, distal appendages of the centriole attract preciliary vesicles (PCVs) released from the Golgi apparatus, which dock at the centriole tip and merge to form mature ciliary vesicles through membrane fusion. This process involves the transformation of the parental centriole into a BB and its distal accessory structures into transition fibers at the base of the cilium [24,25,26]. Concurrently, specific capsid proteins dissociate from the centriole tip, allowing the apical microtubules to extend [24,25,27]. The TZ forms next, serving as a gateway between the BB and the emerging ciliary structure, which subsequently extends to form a ciliary bud [24,25,28]. Finally, the axoneme, dependent on the IFT complex, elongates from the BB tip to complete ciliogenesis [24,25,29]. During this process, the BB is transported from the cell center to the cell surface via kinesin-driven movement within the IFT complex, anchoring at the inner side of the plasma membrane for cilia assembly (Figure 2). Unlike primary cilia, motile cilia rely on the conversion of numerous centrioles into BBs via centriole-dependent and deuterosome-dependent pathways. Notably, motile ciliated cells can generate basal bodies independently of parental centrosomes or deuterosomes, potentially through pericentriolar material (PCM)-associated mechanisms near the nuclear membrane [18,30] (Figure 3). Additionally, the central microtubule of motile cilia extends from the tip of the TZ. Studies indicate that proteins such as *WDR47*, the microtubule-severing enzyme katanin, and members of the Camsap family of microtubule minus-end-binding proteins work together to regulate microtubule formation [31].

## 3. Research on Cilia in Organs and Organ Systems

Current studies have identified over 200 genetic variants associated with ciliopathies and at least 35 distinct cilia-related diseases [12]. These conditions predominantly exhibit hereditary and multisystem involvement. Dysfunction of primitive cilia can lead to situs inversus (mirror-image organ arrangement) [32,33,34]. Abnormal primary cilia are frequently implicated in conditions such as sensory deficits and brain developmental disorders [12,35,36,37,38,39]. By contrast, aberrant motile cilia have been directly linked to primary ciliary dyskinesia (PCD) [40] (Figure 4). As many cilia-related genes regulate ciliary assembly, trafficking, or signaling, their mutations disrupt tissue-specific ciliary functions, resulting in pleiotropic systemic defects.

Cilia research has advanced considerably through next-generation sequencing (NGS), gene editing, super-resolution microscopy (e.g., STED), and cryo-electron microscopy (cryo-EM). These tools enable systematic identification of ciliopathy-associated genes and high-resolution analysis of ciliary ultrastructure. Researchers are now identifying and analyzing cilia-related genes whose mutations are associated with developmental abnormalities and ciliary dysfunction. These efforts are advancing our understanding of the mechanisms underlying numerous disorders. Such investigations have significant theoretical and practical implications for enhancing our knowledge of cilia-related diseases and developing targeted therapeutic strategies.

### 3.1. Role of Cilia in Spermatogenesis

Cilia play key roles in spermatogenesis and meiosis, and the diversity of their functions is reflected in various aspects such as signal transduction, regulation of gene expression, maintenance of cell morphology, and regulation of chromosome dynamics during meiosis. It has been found that cilia show specific expression in different cell types of the testis, such as in Leydig cells, Sertoli cells, and peritubular myoid cells, where the presence of cilia is closely related to cell maturation and function. During spermatogenesis, cilia are involved in signaling pathways in testis cells, which are crucial for the differentiation and function of testis cells, as well as for testis morphogenesis [41,42]. In addition, research has innovatively revealed a novel function of cilia in meiotic recombination, namely, influencing the recombination process by regulating the DNA damage response. Their absence results in compromised DNA double-strand break repair, decreased crossover formation, and elevated germ cell apoptosis [43]. It was also revealed that *Xap5* and *Xap5l* act as antagonistic transcription factors, orchestrating the ciliary transcriptional program during spermatogenesis through direct binding to conserved sequences in ciliary gene promoters [44]. At the same time, the critical role of zygotene cilia in meiotic chromosome bouquets and germ cell morphogenesis has also been demonstrated [45]. Overall, these studies highlight the important role of cilia in spermatogenesis and function, particularly in regulating cell signaling and morphogenesis, and provide new directions for research on male reproductive health.

### 3.2. Sperm Flagellum Structure and Male Infertility

Spermatozoa are structurally specialized reproductive cells responsible for transmitting genetic material to offspring and participating in the fertilization process. They typically comprise two main components: the head and the tail. The tail, or flagellum, is a specialized motile cilium with a complex protein structure that features the characteristic ‘9 + 2’ axoneme configuration [46] found in other motile cilia. However, the sperm flagellum is unique in that its axoneme is surrounded by a dense periaxial structure, except at the distal end [47]. The auxiliary structures around the axoneme are outer dense fibers (ODFs), fibrous sheath (FS), and mitochondrial sheath (MS). The ODF, consisting of nine columns extending from the flagellum’s connecting piece to its main segment [48,49], which provide mechanical support during sperm transport, preventing excessive bending [48,50] and participating in sperm hyperactivation [46]. The fibrous sheath (FS) is another accessory structure that contains glycolytic enzymes capable of providing ATP for flagellar movement [48,49,50,51]. The MS generates energy through oxidative phosphorylation (OXPHOS), providing the energy needed for sperm motility [48].

The sperm flagellum’s intricate structure is essential for motility and function. Any morphological or functional abnormalities can impair sperm motility, resulting in asthenospermia [52] and potentially leading to male infertility. Patients with poor sperm quality often show multidimensional structural defects, and their flagellar system is characterized by a variety of pathological alterations, including structural disorganization of the MSs, abnormalities of the head–tail junction complex, morphological aberrations of the transition zone, aggregation of cytoplasmic membrane remnants, or complete absence of the flagellum [53]. Among them, severe cases, including multiple morphological abnormalities of the flagella (MMAF), manifest as a near-complete hypospermia, with typical morphological alterations that include extreme shortening of the flagellum, flagellar agenesis, and an abnormal conformation of the axoneme helix or an abnormal coiling, etc. [54]. Ultrastructural analysis of MMAF spermatozoa reveals disorganized axonemal and periaxial structures [54], as evidenced by the abnormal attachment of the head of the spermatozoa to a large cytoplasmic vesicle, the spatial disarrangement of microtubule arrays and structural misalignment of the periaxoneme structures seen within it, and the defective structural development of the FSs [53].

### 3.3. Ciliay Function in the Efferent Ductules (EDs)

Following their release from the testis, sperm must undergo migration through the ED, which connects the testis to the epididymis. The outermost ED layer consists of smooth muscle cells, whereas the inner layer comprises ciliated (multiciliated) and non-ciliated epithelial cells [55,56]. The ciliated cells feature tufts of motile cilia with a ‘9 + 2’ structure that exhibit a whip-like, vortex-churning motion. This action agitates the luminal fluid, preventing sperm deposition and blockage [56,57]. By contrast, non-ciliated cells, characterized by apical microvilli, reabsorb luminal fluid to concentrate sperm [58]. These cells are connected via a complex network of tubules, membrane vesicles, and lumen-based microvilli [58]. In the apical region of most non-ciliated epithelial cells, there is a prominent primary ciliated structure whose tip is directly exposed to the luminal fluid environment. This unique spatial distribution feature suggests that this organelle may be involved in physiological regulation as a mechanosensory device, and its function might be related to physiological processes, such as sensing the flow rate or direction of fluid flow in the lumen [56].

EDs play dual roles: facilitating sperm transport and regulating fluid reabsorption, which can concentrate sperm [56,58,59]. Dysfunction of these regulatory mechanisms can lead to fluid accumulation and sperm build-up in the lumen, ultimately resulting in obstructive azoospermia [58,60].

### 3.4. Ependymal Cell (EC) Cilia and Hydrocephalus

In vertebrates, the ventricular system is lined by multiciliated ECs that regulate the cerebrospinal fluid (CSF) flow and circulation [61]. ECs can be classified into two types: multiciliated ventricular cells (E1, with 32–73 cilia, average 49, not internalized in the cell membrane) and biciliated ventricular cells (E2, with only 2 cilia, partially internalized in the cell membrane) [62].

In the physiological activity of ECs, cilia exhibit significant planar polarity, characterized by an asymmetric distribution of basal bodies, a feature that not only regulates the ciliary trajectory but is also a key driver of directional cerebrospinal fluid flow [62]. ECs also demonstrate a coordinated pattern of arrangement at the multicellular facet, a supracellular-level synergistic mechanism that ensures the spatiotemporal coherence of ciliary pulsation within a localized area [63]. Hydrocephalus, abnormal dilatation of the ventricular system, and its pathomechanism involve disturbances in cerebrospinal fluid production, dynamic homeostasis, and metabolic processes [64], which are maintained in homeostatic equilibrium under normal physiological conditions by a sophisticated regulatory network [65,66]. Many gene loci associated with hydrocephalus have been identified [66], among which are genes involved in the regulation of ciliary structure and function. For instance, *Wdr78* is involved in axonemal power arm assembly; its absence leads to dysfunctional ciliary motility and impaired cerebrospinal fluid flow, in turn leading to hydrocephalus [67]. Both primary and motile cilia are essential for EC planar polarity [61]. Dysfunction in either type of cilia can disrupt planar polarity, ultimately contributing to hydrocephalus.

### 3.5. Neuronal Cilia and Nervous System Diseases

Neuronal cilia are an important cellular structure in the nervous system, and recent studies have revealed their critical role in neuronal function, development, and disease. Neuronal cilia are categorized into primary cilia and motile cilia, mainly primary cilia, which are possessed by most central nervous systems [68,69]. Cilia ultrastructure, localization, and orientation vary at different developmental stages and cell types, reflecting the high diversity of cilia in the nervous system [70]. Many developmental and degenerative genetic disorders, such as Joubert syndrome and Meckel syndrome, are associated with ciliopathies characterized by primary cilia structural and length abnormalities [71]. Numerous studies have shown that abnormalities in the function of primary cilia in nerve cells are often associated with intellectual disability, neurodevelopmental abnormalities, and even cancer [72,73]. Abnormalities in the function of receptors and kinases specifically expressed in cilia are strongly associated with the development of these neurological disorders. Primary cilia regulate neural precursor cell proliferation and differentiation [74], mediating Sonic hedgehog (Shh) signaling [74,75], as well as regulating astrocytes’ development [76,77]. Neuronal primary cilia are involved in hippocampal memory by influencing adult neurogenesis [78], and forebrain-specific removal of primary cilia affects emotional experience and behavioral responses [79]. Hypothalamic cilia may be involved in appetite regulation [80,81]. Neuronal cilia serve as specialized signaling platforms that play crucial roles in physiological and pathological processes within the nervous system. In-depth analysis of the molecular mechanisms of cilia will provide important clues for understanding the complexity of the nervous system and developing new therapeutic strategies.

## 4. Advancements in Cilia Imaging Technology

### 4.1. Optical Imaging

#### 4.1.1. Conventional Optical Imaging

Advances in microscopy have considerably enhanced our understanding of cilia structure and function. Cilia were first observed in the 17th century when Antonie van Leeuwenhoek used an early optical microscope to study protozoan motile cilia. He described them as ‘little legs or feet that can move fast’ [82,83], offering the first account of their motor function. Pioneering work on cilia dates back to the early 1830s, when Purkinje and Valentin first observed motile cilia in mammalian tissue samples, a discovery that marked the beginning of the study of the biology of motile cilia [84]. The exploration in this period mainly relied on optical microscopy, and the objects mostly focused on the locomotor organs of *Protozoa* and male gametes, etc. [82]. The discovery of non-motile primary cilia coincided with studies on centrosomes, centrioles, and BBs. Centrosomes were first described by Flemming in 1875 and Van Beneden in 1876 [84]. However, the resolution limits of traditional optical microscopy precluded detailed observation of cilia and related structures, restricting early progress in cilia research [85].

Optical coherence tomography (OCT), an advanced imaging technique that uses near-infrared light as a light source to obtain cross-sectional or three-dimensional images of tissue by measuring the interference and scattering signals of the light within the sample, enables non-invasive, contact-free tomographic imaging of biological tissues [86]. Larina’s research group used OCT to visualize ciliary metachronal wave propagation in mouse oviducts, providing new insights into female reproductive physiology [87] (Figure 5). The discovery of green fluorescent proteins (GFPs) and other spectral fluorescent proteins has revolutionized cilia imaging by enabling simultaneous molecular and dynamic observations of biological samples [88,89]. Furthermore, fluorescent probes have been extensively employed to enhance image quality, often in combination with advanced labeling techniques. Imaging modalities such as fluorescence, bioluminescence, and Raman imaging are indispensable for acquiring non-invasive two-dimensional or multidimensional image data at multiple spatial scales [90]. Fluorescent labeling has proven particularly valuable in this field, enabling researchers to focus on ciliary axonemes, basal bodies, or associated signaling complexes, evaluate cilia morphology, monitor ciliogenesis, and correlate ciliary dynamics with gene expression patterns using in situ hybridization. These advancements have considerably deepened our understanding of cilia structure, function, and regulatory mechanisms.

Fluorescent proteins such as GFP, emerald, yellow fluorescent protein (YFP), mCherry, mCerulean3, TagRFP, and TdTomato are widely used for direct visualization of ciliary structures. Non-fluorescent self-tagged proteins like HaloTag, SNAPtag, and CLIP-tag enable indirect labeling through the binding of fluorescent ligands [91,92,93,94,95,96]. For instance, Hwang’s research group used imaging techniques to analyze the co-expression of multiple ciliary markers, identifying the sequential events of ciliogenesis in developing cilia [97]. Similarly, fluorescent tagging enables real-time monitoring of IFT molecule movement, thereby elucidating IFT’s critical role in constructing ciliary structures and functions [84,98,99]. Moreover, Pan et al. have developed innovative tubulin staining techniques that offer improved resolution and reduced background noise, opening new avenues for studying cilia morphology [100] (Figure 6).

The integration of genetically encoded self-labeling tags with structurally optimized synthetic dyes has further enhanced quantum efficiency and photon yield by improving brightness and photostability, thereby advancing cilia imaging methods [101]. While most live-cell imaging of mammalian cilia relies on exogenous overexpression of the fluorescent markers, the advent of CRISPR/Cas9 technology now enables the knock-in of fluorescent reporter genes into endogenous loci. This breakthrough facilitates real-time imaging of ciliary proteins at physiologically relevant expression levels, avoiding artifacts caused by overexpression [102].

Nonetheless, several critical factors must be addressed during research. Artifacts associated with overexpression, the impact of fluorescent labels on protein function, and the intrinsic fluorescence of cells are key considerations. Dye-specific properties, such as solubility, toxicity, stability, aggregation tendencies, and cell permeability, must be carefully evaluated. Additionally, the localization of fluorescently labeled proteins should align with that of their endogenous counterparts. To minimize interference with cellular functions, fluorescently labeled proteins should be expressed at levels comparable to endogenous proteins [103,104]. For high-resolution analysis, electron microscopy (EM) remains essential due to its nanometer-scale resolution, which is critical for accurately identifying ciliary structures. Researchers typically use optical imaging with specific labeling techniques as a preparatory step before transitioning to EM [105].

#### 4.1.2. Optical Imaging Technology: Pushing the Limits of Conventional Optical Diffraction

The direct observation of cilia using fluorescent probes is limited due to the inherent constraints of traditional optical imaging techniques, which prevent clear visualization of nanoscale ciliary structures. Therefore, it is crucial to use more advanced methods, such as super-resolution fluorescence imaging, to visualize proteins at the single-molecule level and on the nanoscale. This breakthrough has allowed us to overcome previous barriers and investigate the structure and dynamic activities of cells with unprecedented spatial and temporal resolution [106,107]. The development of structured illumination microscopy (SIM, achieving ~50–100 nm resolution), stimulated emission depletion microscopy (STED), and single-molecule localization microscopy (SMLM) has enabled the detailed documentation of the complex structures and dynamic processes involved in ciliogenesis and transport [108,109,110] (Figure 7). Li’s research group used three-dimensional SIM to gain valuable insights into the location of kinesins on the motile cilia axonemes. Furthermore, they utilized grazing incidence structured illumination microscopy (GI-SIM) to reveal differences in protein behavior [111] (Figure 8).

Although SIM has greatly enhanced our understanding of cilia, further exploration using techniques such as cryo-EM is still required to examine their ultrastructure more comprehensively.

### 4.2. Electron Microscopy (EM) Imaging

#### 4.2.1. Transmission EM (TEM)

The advent of EM has effectively addressed the challenge of resolution. TEM’s superior resolution, high magnification, and wide dynamic range have solidified its role as a cornerstone in cellular and tissue research [112]. Over time, TEM has seen increasing application in cilia research, particularly in the analysis of ciliary structure, and function, as well as the molecular mechanisms underlying associated diseases. Visualization of cilia-related structures using TEM remains a reliable benchmark in this field. The use of TEM in cilia research dates back to 1946 [113], when Jakus and Hall [114] employed TEM to observe the cilia of *Paramecium*. It was not until 1954 that TEM became widely adopted in cilia studies. Fawcett’s seminal study of ciliated epithelial cells revealed and classified the ultrastructure of mammalian cilia [115], considerably advancing the understanding of ciliary structures and providing early experimental evidence supporting the microtubule sliding theory [116].

Historical advancements in cilia research further shaped the field. Around the late 19th century, Boveri introduced the concepts of the ‘centrosome’ and ‘centriole’ [84], which expanded knowledge of cilia and their associated components [84]. Engelmarm was the first to identify and name the ‘BB’ [84], which, alongside the axoneme and centriole, formed the foundational structural model of cilia [85]. Zimmermann later observed non-motile cilia with paired centrioles in mammals, distinguishing them from the motile cilia’s ‘central flagellum’ [84]. Sorokin utilized TEM to examine ciliogenesis in rat lungs, coining the term ‘primary cilia’ to describe these structures [85].

In the 1990s, Kozminski’s research used video-enhanced differential interference contrast microscopy to observe bidirectional protein transport in the flagella of *Chlamydomonas reinhardtii*, introducing the concept of ‘IFT’ [12,13]. Porter demonstrated that motile cilia have a ‘9 + 2’ axoneme structure, while primary cilia have a ‘9 + 0’ structure [117]. Extensive research employing TEM has documented changes in cilia under various physiological and pathological conditions, elucidating their functional roles and implications [12]. In clinical settings, TEM has become a key diagnostic tool for identifying ultrastructural abnormalities in cilia. It is considered the gold standard for diagnosing PCD. Notably, even in cases where ciliary ultrastructure appears normal, PCD can still be diagnosed based on clinical symptoms and genetic findings, accounting for approximately 30% of cases. Although the reliance on TEM for PCD diagnosis has decreased with the advent of advanced imaging techniques [118], it remains invaluable for identifying ultrastructural defects. This capability supports the development of tailored treatment strategies based on specific ciliary defects.

#### 4.2.2. Cryo-EM

Cryo-EM, which was awarded the Nobel Prize in Chemistry in 2017 for advancements in structural biology, is a groundbreaking technique that enables the three-dimensional reconstruction of biological macromolecules [119,120]. This method involves freezing materials at ultra-low temperatures, facilitating the observation and analysis of biological samples while allowing high-resolution imaging at the atomic level [121] (Figure 9). Cryo-EM employs two primary approaches for imaging: cryo-electron tomography (cryo-ET) and single-particle analytical imaging.

The structural data produced by these techniques have considerably advanced our understanding of the mechanisms underlying ciliary movement, particularly the action mechanisms of kinesin and their role in ciliary dyskinesia [124]. In 2006, Nicastro et al. performed a three-dimensional reconstruction of motility cilia duplex microtubules from *Chlamydomonas reinhardtii* using electron Cryo-ET. This study revealed that various protein structures were arranged in a periodic pattern at the periphery of the duplex microtubules, including inner wall dynein (IDA), outer wall dynein (ODA), and microtubule-interacting proteins, which were strongly linked within the microtubules [125]. In addition to duplex microtubule structures, Walton’s research investigated the structural features of the RS, ODA, IDA, and other complexes at near-atomic resolution [126]. The integration of microcryo-electron imaging of kinesin polymorphisms within the molecular structure of cilia, coupled with the localization of subunits in the microtubule junctional protein–kinesin regulatory complex using SNAP-tag protein tagging, has provided critical insights into axonemal kinesin regulators, which are vital for studying cilia and flagellar motility [127]. A study used cryo-EM with single particles to reveal the near-atomic-level resolution structure of ciliated microtubules in *Chlamydomonas reinhardtii*. The study identified 33 intramicrotubule proteins, and an atomic model was developed [128] (Figure 10). 

From 2021 to 2022, the architectures of duplex microtubules in animal and human respiratory cilia were resolved with near-atomic precision, highlighting both the evolutionary conservation and variation of proteins within the microtubules [129,130]. Sun et al. determined the molecular structure of the cytoskeleton (axoneme) in mammalian primary cilia with cryo-ET. Their findings revealed that the ‘9 + 0’ structure is an oversimplification, applicable only to the ciliary base, highlighting the need for nuanced models of ciliary architecture. This discovery paved the way for future research into the structure and function of cilia [131] (Figure 11).

These findings open new avenues for enhancing our understanding of cilia at the molecular level and for conducting accurate, efficient studies on cilia gene activity. However, cryo-EM remains a complex and evolving field, requiring further exploration and development. The procedures and equipment used in cryo-EM demand expertise from multiple disciplines, and it is likely that certain unknown factors or technical challenges will influence sample preparation and, consequently, the experimental results. Moreover, different types of experimental equipment can yield varying results [132].

#### 4.2.3. Volume EM (vEM)

While electron microscopes offer exceptional insights into ultrastructure, traditional two-dimensional imaging lacks depth perception, limiting our understanding of the three-dimensional volumes, dimensions, and other structural properties of living organisms. However, volumetric EM techniques allow for the acquisition of successive EM images, enabling the creation of precise three-dimensional models and high-resolution, large-volume three-dimensional images of cells or tissues. These images facilitate the examination of organelles within their cellular context [133,134,135,136,137,138] (Figure 12). This technology has been recognized as one of the ‘7 technologies to watch in 2023’ by Nature [139].

vEM has been used for morphological studies of cilia. Wang’s group used focused ion beam scanning electron microscopy (FIB-SEM) to image and three-dimensionally reconstruct the multicilia of ventricular membrane cells and investigated at the ultrastructural level how *Cenp*j mutations induced primary microcephaly [140]. Mytlis et al. used TEM, serial block-face scanning electron microscopy (SBF-SEM), and other techniques to discover the presence of primary cilia in the meiotic prophase of zebrafish primary oocytes [45]. Meanwhile, Sheu et al. discovered primary cilia on the surface of a neuron, which formed a specialized connection with the axon of another neuron (called axo-ciliary synapse), through [141] (Figure 13). Another study analyzed cilia in human pancreatic islets and determined their size and formation with FIB-SEM [142]. As previously established, substantial research has focused on the intrinsic structure of cilia. vEM can observe the surface contours and structures of samples at exceptional resolution. This allowed for observation and measurement of morphological features, including axon length and diameter.

Advancements in sophisticated imaging technologies have made it easier to obtain atomic-level images of the entire cilia. However, much remains to be understood about the molecular principles driving ciliary movement. To fully understand these processes, we must investigate the complex structural dynamics of cilia during different oscillatory phases.

## 5. Conclusions

Cilia, with their unique structure and functional characteristics, have increasingly become a focal point in the biomedical sciences, especially in understanding cellular processes and diseases. Advanced imaging methods, such as EM and cryo-EM, have enabled detailed examination of the complex ultrastructure of cilia. Clarifying the structure of cilia and the intricate alterations within them necessitates the continuous development and application of sophisticated imaging techniques. To fully understand ciliary movements, a combination of traditional and advanced imaging methods is essential. While studies of cilia in the reproductive system remain limited, particularly in the male reproductive system, further understanding of cilia will facilitate their biomedical application.

## Figures and Tables

**Figure 1 biology-14-00521-f001:**
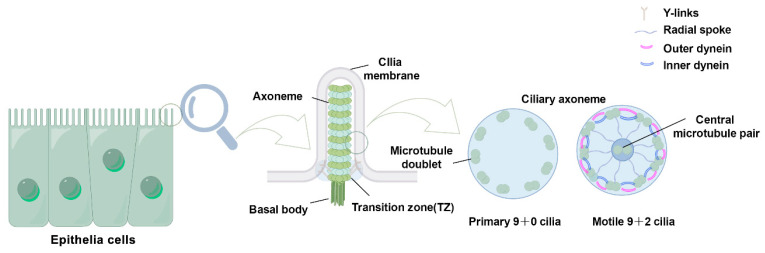
Schematic structure of the cilia. Cilia possess four primary structural elements: the basal body, axoneme, transition zone, and ciliary membrane. The transition zone is composed of Y-link structures that connect the microtubules to the ciliary membrane. Transition fibers connect the basal body to the plasma membrane. Cilia are classified according to their microtubule configuration: primary cilia have a ‘9 + 0’ microtubule arrangement (9 pairs of double microtubules), whereas motile cilia have a ‘9 + 2’ one (additional central pairs) and include ciliary movement-associated structures, such as inner dynein arms, outer dynein arms, and radial spokes (RS).

**Figure 2 biology-14-00521-f002:**
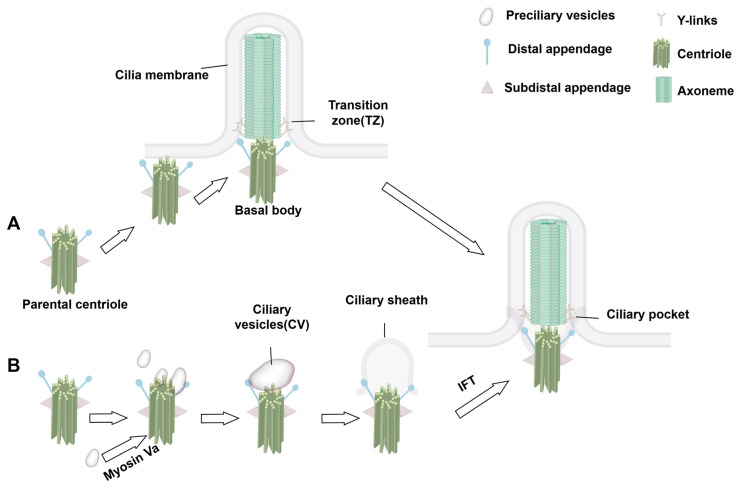
Initiation process of ciliogenesis. (**A**) Extracellular pathway: The mother centriole migrates to the cytoplasmic membrane through the mediation of its distal appendages and becomes anchored, followed by the initiation of axoneme assembly. (**B**) This process occurs within the cytoplasmic compartment prior to the formation of the axoneme and ciliary membrane. Preciliary vesicles (PCVs), dependent on myosin Va-mediated transport, are specifically recruited to the distal appendage of the mother centriole. These vesicles undergo coordinated membrane fusion events to generate functional ciliary vesicles (CVs). During the sheath development phase, synchronized development of the axoneme and ciliary membrane is achieved.

**Figure 3 biology-14-00521-f003:**
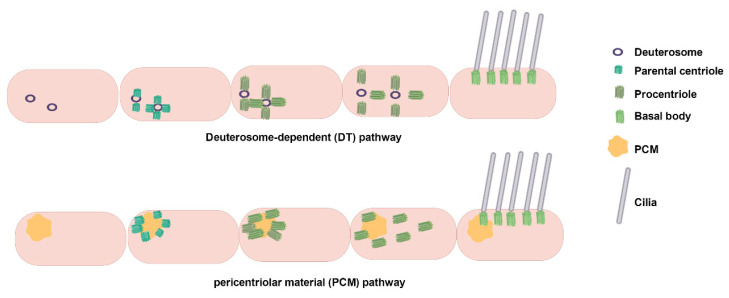
Multiciliated cell basal body production pathways: Deuterosome-dependent and PCM-dependent pathways. A deuterosome is a unique structure of multiciliated cells during the centriole amplification stage and disappears when amplification is completed. The deuterosome-dependent pathway mediates the formation of procentrioles directly, subsequently aggregates them for maturation, and ultimately releases them into the cytoplasm. The deuterosome pathway is the main mechanism of centriole generation in multiciliated cells, and about 90% of centrioles in mammalian multiciliated cells are generated through this pathway. Pericentrin, a PCM component, can recruit centromeric proteins, thereby promoting centromerogenesis. Abbreviation: PCM, pericentriolar material. Reproduced from Ref. [30]. Copyright 2020, Trends in cell biology.

**Figure 4 biology-14-00521-f004:**
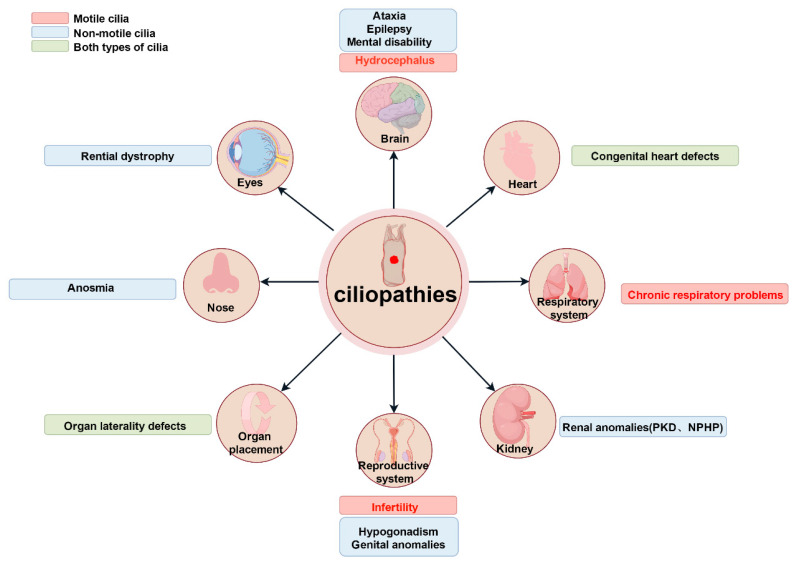
Ciliopathies impacting human organ systems due to dysfunction of motile and non-motile cilia. The illustrations show the different ciliopathies of tissues or organ systems and highlight the key manifestations in each affected organ. Red, blue, and green to show conditions from motile cilia issues, primary cilia defects, and both types of abnormalities. Abbreviations: PKD, polycystic kidney disease; NPHP, nephronophthisis. Reproduced from Ref. [12]. Copyright 2017, Springer Nature.

**Figure 5 biology-14-00521-f005:**
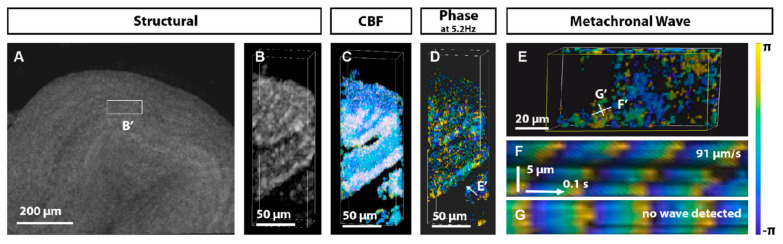
3D OCT analysis of cilia metachronal activity in the mouse oviduct. (**A**) 3D OCT visualization provides a holistic view of the tubal jugular ampulla and designates specific regions for 3D+time imaging. (**B**) Structural OCT volumetric perspective of selected regions reveals ciliated oviductal grooves. (**C**) Corresponding frequency mapping. (**D**) 5.2 Hz phase diagram of the dominant frequency. (**E**) Viewing the phase distribution on the groove surface from the perspective shown in (**D**). (**F**) Plotting the change in phase over time along the line marked in plot E shows the metachronal wavefront propagation at 91 μm/s. (**G**) In the orthogonal orientation, no discernible waves are observed. Reproduction from Ref. [87].

**Figure 6 biology-14-00521-f006:**
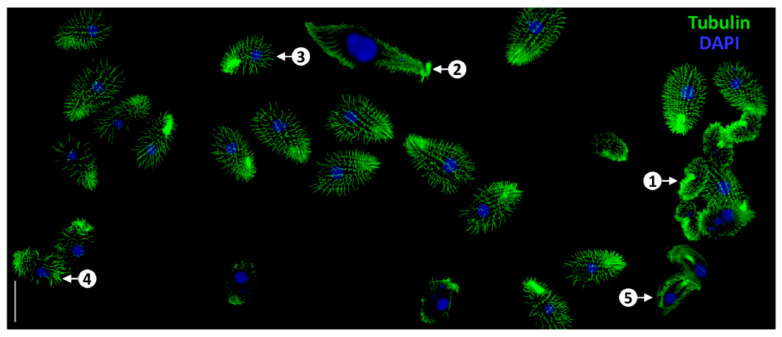
Successful outcome of tubulin staining in a mixed population of ciliate cells, including *C. inflata* (1), *Lacrymaria olor* (2), *T. thermophila* (3), *C. hirtus* (4), and *C. uncinata* (5). Scale bars = 50 μm. Reproduction from Ref. [100]. Used under Creative Commons CC-BY license.

**Figure 7 biology-14-00521-f007:**
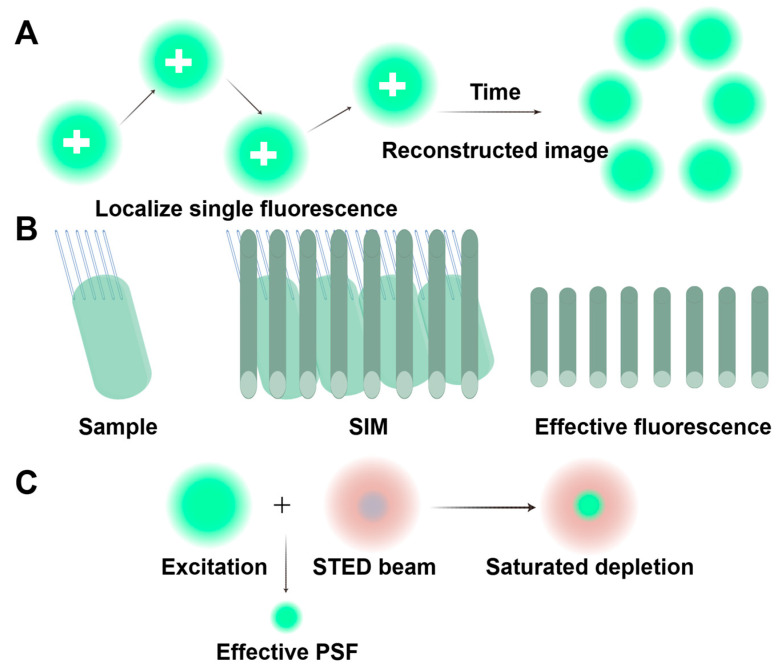
Principles of the three main super-resolution techniques. (**A**) The principle of single-molecule localization microscopy (SMLM). SMLM activates and precisely locates individual fluorescent molecules in repeated cycles. By compiling these isolated positions, it constructs a 3D super-resolution image that surpasses conventional resolution limits. (**B**) The principle of structured illumination microscopy (SIM). SIM uses patterned light to create low-frequency interference (Moiré pattern) with samples. By capturing shifted images, it reconstructs observable structures, but details finer than the pattern’s frequency (like the upper part of the sample) remain undetectable. (**C**) The principle of stimulated emission depletion (STED). STED microscopy uses a donut-shaped laser (center-aligned with the excitation spot) to suppress peripheral fluorescence via a saturation effect and narrows the emission area below the diffraction limit, enabling super-resolution imaging. Reproduced from Ref. [107], Chen, J., et al. Used under Creative Commons CC-BY license.

**Figure 8 biology-14-00521-f008:**
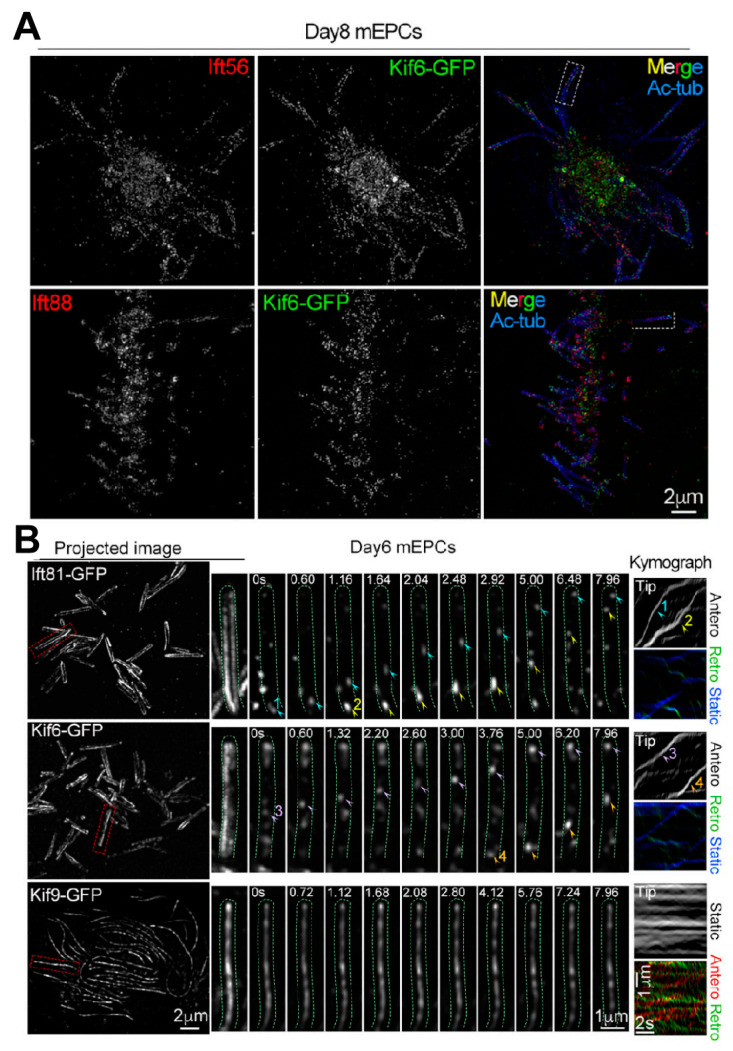
Spatiotemporal dynamics of exogenous Kif6/Kif9 in ependymal cilia. (**A**) 3D SIM visualization revealed spatial distributions of Kif6-GFP relative to Ift56 and Ift88, with acetylated tubulin (Ac-tub) serving as axonemal markers. (**B**) GI-SIM dynamic imaging captured motile trajectories of ciliary Kif6-GFP and Kif9-GFP. The mEPCs underwent adenoviral transduction to express GFP-tagged constructs, with Ift81-GFP as a positive control. Arrowheads highlight traceable GFP puncta demonstrating directional movement patterns. The trafficking trajectories of the GFP puncta projected in the first 200 frames of videos 1, 2, 3 and 4 are shown as projected images. From Ref. [111]. Used under Creative Commons CC-BY license.

**Figure 9 biology-14-00521-f009:**
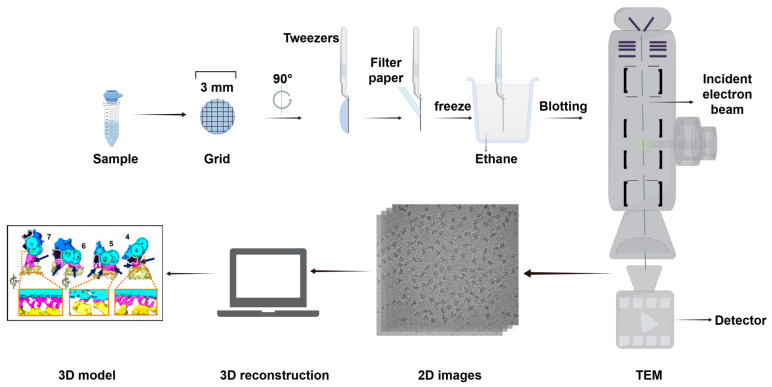
Basic steps in structural analysis by cryo-EM. The basic process consists of several basic steps, such as sample preparation, rapid freezing of samples, TEM imaging, image processing, and structural analysis. Abbreviation: TEM, transmission EM. Reproduced from Ref. [122]. Copyright 2016, Springer Nature. 3D model from Ref. [123], Hughes et al. Used under Creative Commons CC-BY license.

**Figure 10 biology-14-00521-f010:**
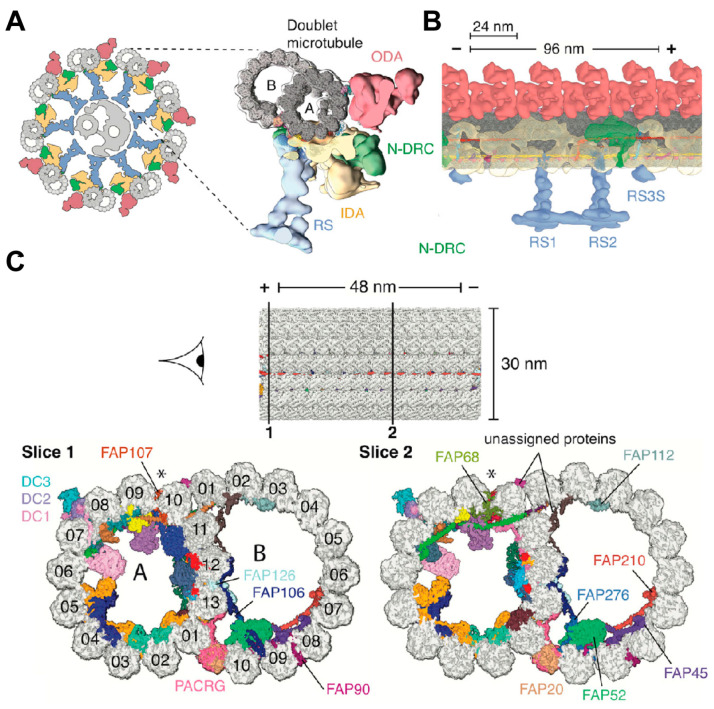
Cryo-EM image of axonemal doublet microtubule-associated structures. (**A**) Left: Schematic of *C. reinhardtii* axoneme: nine doublet microtubule encircling central pair (gray), with RS (blue), IDA (yellow), N-DRC (green), and ODA (red). (**B**) Longitudinal Cryo-EM image of the 96 nm repeat of doublet microtubule structures docked into the subtomogram of the axoneme. (**C**) Cross-sections showing color-coded microtubule inner proteins (MIPs). The minus (-) and plus (+) tips of the microtubules are labeled in the figure, and the asterisks indicate the seams of the A microtubules. From Ref. [128]. Copyright 2019, Cell.

**Figure 11 biology-14-00521-f011:**
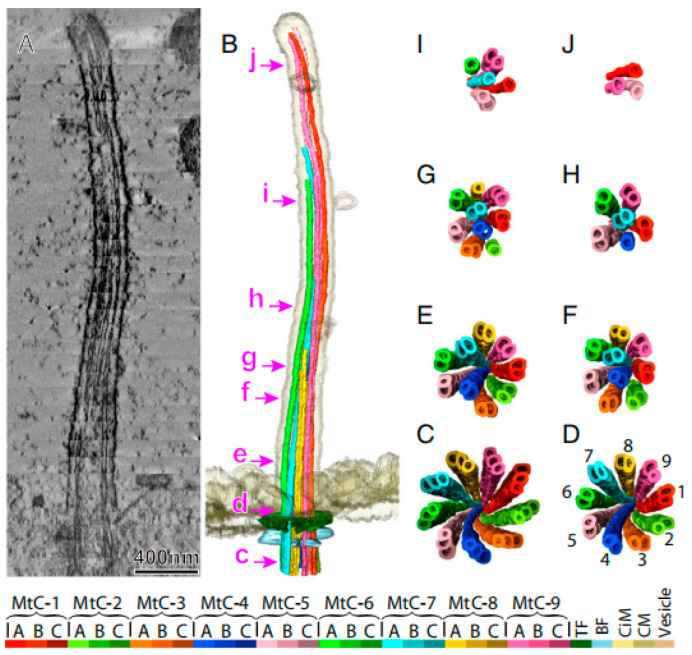
Primary cilia ultrastructure. A 3D model of primary cilia reconstructed from 33 dual-axis tomography datasets. (**A**) Central longitudinal slice (13.4 nm thick). (**B**) Amira software-based structural model. Panels C–J show microtubule changes along the cilium (view: base to tip), corresponding to the positions labeled in (**B**). Color-coded structures (see legend). Abbreviations: BF (basal foot), CiM (ciliary membrane), CM (cytoplasmic membrane), MtC-A/B/C (microtubule complex A/B/C-tubule), and TF (transition fiber). From Ref. [131]. Copyright 2019, Proc. Natl. Acad. Sci.

**Figure 12 biology-14-00521-f012:**
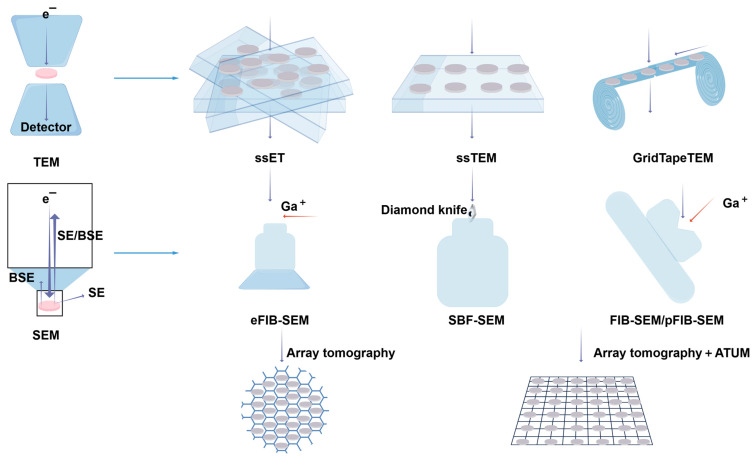
Overview of vEM techniques. vEM integrates TEM- and SEM-based methods for 3D ultrastructural imaging. vEM employs electron beam–sample interactions to generate serial images: TEM detects transmitted/scattered electrons through ultrathin sections (50–70 nm), while SEM captures backscattered/secondary electrons (BSE/SE) from surfaces. TEM modalities include serial-section electron tomography (ssET) reconstructing 200–300 nm slices via tilt-series imaging, serial-section TEM (ssTEM) for grid-mounted sections, and GridTape TEM automating high-throughput tape-fed imaging. SEM methods involve eFIB-SEM (enhanced ion milling with bias control), SBF-SEM (diamond knife sectioning), FIB-SEM/pFIB-SEM (ion/plasma beam milling), or sequential imaging of serial slices on a substrate such as a silicon wafer (array tomography). Ultramicrodissection is partially automated by collecting the slices on a tape. Abbreviations: FIB-SEM, focused ion beam SEM; SBF-SEM, serial block-face scanning electron microscopy; ATUM, automatic tape-collecting ultramicrotome. Reproduced from Ref. [138], Peddie, C. J., et al. Used under Creative Commons CC-BY license.

**Figure 13 biology-14-00521-f013:**
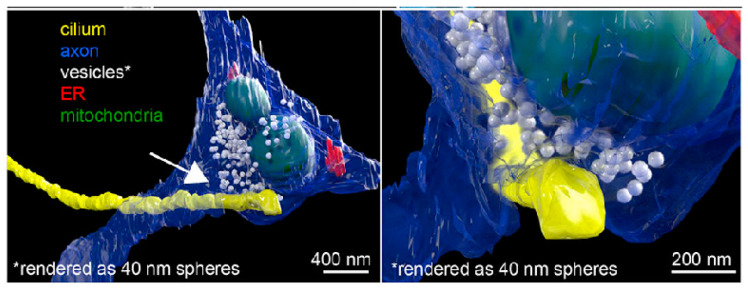
FIB-SEM reveals axo-ciliary synapses. From Ref. [141]. Used under Creative Commons CC-BY license.

## Data Availability

No data were used for the research described in the article.

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
