# Peer review of "Advances in Imaging Techniques for Mammalian/Human Ciliated Cell’s Cilia: Insights into Structure, Function, and Dynamics"

_biology, 2025, doi:10.3390/biology14050521_

Round 1
Reviewer 1 Report (Previous Reviewer 1)
Comments and Suggestions for Authors
I would suggest accepting the paper in current form as is. My original concerns are addressed now. My only comment is:
Data Availability Statement: Data are contained within the article.
A review paper is not supposed to contain experimental data. Please correct
Author Response
Comments 1: I would suggest accepting the paper in current form as is. My original concerns are addressed now. My only comment is: Data Availability Statement: Data are contained within the article. A review paper is not supposed to contain experimental data. Please correct.
Response 1: Thank you very much for our kind comments. The statement was corrected accordingly.
Reviewer 2 Report (Previous Reviewer 2)
Comments and Suggestions for Authors
Dear authors,
Thank you for revising the manuscript based on my recommendations.
In general, I am satisfied with your answers. However, I still insist that Fig. 9 should show an electron microscope, not a simple light microscope! I frankly do not understand why this is difficult to do in a schematic drawing! Yes, the authors changed the picture, but it is still a light microscope, not an electron one. Even if a simple cartoon diagram is given, it should not mislead the reader of a scientific paper.
The authors now reproduce the figures from the original articles that illustrate the advantage of a particular method in the study of cilia. I fully agree that, in general, such drawings are extremely important and necessary. However, it seems to me that the review in its current form has become too overloaded. It was not necessary to reproduce the figures in their entirety, but to take only the most illustrative, most representative figures.
Author Response
Comment 1: Dear authors, Thank you for revising the manuscript based on my recommendations. In general, I am satisfied with your answers. However, I still insist that Fig. 9 should show an electron microscope, not a simple light microscope! I frankly do not understand why this is difficult to do in a schematic drawing! Yes, the authors changed the picture, but it is still a light microscope, not an electron one. Even if a simple cartoon diagram is given, it should not mislead the reader of a scientific paper. The authors now reproduce the figures from the original articles that illustrate the advantage of a particular method in the study of cilia. I fully agree that, in general, such drawings are extremely important and necessary. However, it seems to me that the review in its current form has become too overloaded. It was not necessary to reproduce the figures in their entirety, but to take only the most illustrative, most representative figures.
Response 1: Thank you very much for your valuable comments. We sincerely apologize for the oversight regarding Figure 9. We have redrawn this figure in the revised manuscript to accurately depict an electron microscope, ensuring clarity and preventing any potential misunderstanding for our readers. The updated schematic is included below and in the revised manuscript as Figure 9.
Regarding your concern about the reproduced figures, we initially added these based on another reviewer's request during the last round of revisions. However, we recognize the importance of balance and clarity in visual content. To address your concerns, we have simplified Figures 8 and 13 to retain only the most illustrative and representative images. These revisions aim to support the article's arguments effectively while enhancing reader comprehension without overwhelming them with excessive detail. Thank you again for your constructive comments and support (Please see the attachment).

Reviewer 3 Report (Previous Reviewer 3)
Comments and Suggestions for Authors
The authors have made all revisions. I have no more comment.
Author Response
Cooment 1: The authors have made all revisions. I have no more comment.
Response 1: Thank you very much for your valuable comments.
This manuscript is a resubmission of an earlier submission. The following is a list of the peer review reports and author responses from that submission.
Round 1
Reviewer 1 Report
Comments and Suggestions for Authors
The topic of this review is interesting and falls within the scope of the journal. However, I cannot recommend its publication in the current form, because it lacks illustrative material. There are only 4 schemes (Fig 1-4), however one would expect to have other images, showing the cilia and its functions. For example, the whole chapter 4. Advancements in cilia imaging technology does not feature a single microscopy image, which is definitely weird. The authors are requested to select appropriate previously published images, obtain the permission to republish and arrange them into figures. After that, the paper can be re-reviewed.
Author Response
Reviewer 1
The topic of this review is interesting and falls within the scope of the journal. However, I cannot recommend its publication in the current form, because it lacks illustrative material. There are only 4 schemes (Fig 1-4), however one would expect to have other images, showing the cilia and its functions. For example, the whole chapter 4. Advancements in cilia imaging technology does not feature a single microscopy image, which is definitely weird. The authors are requested to select appropriate previously published images, obtain the permission to republish and arrange them into figures. After that, the paper can be re-reviewed.
Response to Reviewer: We sincerely appreciate your valuable comments. Given that the editor has allowed only 10 days for the revision of our manuscript, we find this timeframe inadequate for securing the necessary permissions for the published images from both the original authors and the journals involved.
To respond to the reviewer’s comments, we have drawn several new illustrations (Figure 5-7) to improve the readability of our manuscript.
Reviewer 2 Report
Comments and Suggestions for Authors
This review focuses on cilia and flagella as some of the most amazing and intriguingly specialized parts of the cell. These organelles characterize a wide and diverse range of cells that specialize in performing motor, sensory, locomotor, and many other important functions. The ciliated cells appeared very early in evolution; flagella also appeared long ago. For example, some protists have flagella, such as Chlamydomonas, which serves as a traditional model object for their study, as well as spermatozoa, the most famous flagellated cells that appeared along with sexual reproduction. Although cilia/flagella were first observed as early as in the 18th century, ciliary biology is still actively developing. New research perspectives that were until recently unthinkable are opening up with the development of high-resolution ultramicroscopy techniques, especially on living cells, as well as approaches that allow the structure of cilia/flagella to be assessed at the level of atomic resolution.
In the first part of the review, its authors note the fundamental principles of the structure and functions of cilia, and at the end they place special emphasis on modern achievements in ciliary biology using the latest imaging technologies.
Overall, I liked this review. It was written briefly, but at the same time capaciously, being easy and interesting to read. At the same time, I would like to address the authors with some comments/suggestions that, in my opinion, could improve the review, making it even more valuable and attractive to readers.
- Although the review mentions, for example, Chlamydomonas as a model system for studying cilia and flagella, it actually focuses mainly on biomedical applications, including human ciliary pathologies. In this regard, I think that the authors should either specify the title, limitingittomammalian/humanciliatedcells, orexpandthereviewby adding a relevant section(s) on the diversity of flagella in eukaryotes and approaches to their study.The first is easier to do, the second would be more interesting.
For example, the review does not discuss the evolutionary aspect at all. In some groups of organisms, it is the structure of sperm flagella that allows us to solve evolutionary-taxonomic questions, determine the mono- or polyphyletic origin of a group, and serves as a diagnostic criterion in taxonomy. If the title of the article is left as is, in a broad sense, then the authors' assertion that only motile sperm flagella have a 9 + 2 axoneme structure (e.g., lines 56–57; 179–180) is not entirely accurate. For example, not only the classic 9+2 pattern, but also the 9+0 structure and the specific 9+'1' trepaxonematan pattern characterize the sperm of flatworms. A bulk of work is devoted to this topic, but basically, as far as I know, they are carried out using conventional transmission electron microscopy (TEM). Another interesting example is the recent discovery of flagella in some amoeboid protists. Are modern imaging techniques used in this area and how can they help in solving issues of zoology, systematics and phylogeny?
- Of course, the review should include a discussion of publications on the topic from the last year or two. There are such works, including several reviews. For example, I think it would be worth mentioning the neuronal cilium, which would further attract the attention of readers with an emphasis on recent research.
- In my opinion, the review would have sparkled with completely different colors if its part devoted to the application of modern microscopy methods and achievements to study cilia/flagella had been supplemented with appropriate illustrations. I understand that authors may face permission issues, but the reader would have been able to immediately evaluate the advantages of this or that modern visualization method without referring to the original source.
- Regarding the methods used to study the fine structure of cilia, I would like to ask the authors whether, for example, immunoelectron microscopy (IEM), high-voltage electron microscopy, atomic force microscopy, including in vivo imaging, as well as correlative microscopy, which allows the structure of the same cell to be determined simultaneously at the light and electron level, are used for this purpose, and if so, to what extent?
- Important note. Section 4 needs to be restructured/rewritten.
The Section 4.4. “Optical imaging technology” should be moved into the Section 4.1 “Optical imaging”.
Similarly, aren’t cryo-EM (Sect. 4.3) or volume EM/ET (Sect 4.5) parts of the more extensive and diverse field of electron microscopy (Sect. 4.2)?
Therefore, I would suggest that the authors divide Section 4 into two large sections — Optical imaging and Electron microscopy imaging — with several relevant subsections within.
Finally, in addition to the above, lines 336–342 seem out of place in the Section “Electron microscopy”, since Ref. [94] (Rix et al., 2011) contains no EM data.
Additional minor points
- Lines 89–91: The cited work [21] is not a “recent” study, it appeared in 2014.
- Line 120: The abbreviation PCM should be expanded upon when it is first mentioned here, and not only in Fig. 3 and in the caption to it.
- Lines 269–270: For a wide range of readers it would be useful to briefly explain the essence of the methodOCT.
- Line 323: The name of the genus Paramecium should be italicized.
- Please check your in-text citations, as it is customary to cite a work by its first author, not its last.If the author is the last one in the publication, you can write: “Someone’s research group”, but not “Someone et al.” This can be misleading.Below are some of the citation inconsistencies I have found:
Line 392: Ref. 110 is Fang et al., not Yan et al.
Line 416: Yao Cong et al. is not in the Reference list.
Lines 417–422: Gaia et al. is not in the Reference list. However, the sentence (“This discovery...”) ends with the reference [117] to the work by Kiesel et al.
Line 426: Hughes is the last author of the work Polino et al. [119].
- Electron microscopy does not deal with living cells (cf. line 401), only with fixed ones.
Author Response
Reviewer 2
This review focuses on cilia and flagella as some of the most amazing and intriguingly specialized parts of the cell. These organelles characterize a wide and diverse range of cells that specialize in performing motor, sensory, locomotor, and many other important functions. The ciliated cells appeared very early in evolution; flagella also appeared long ago. For example, some protists have flagella, such as Chlamydomonas, which serves as a traditional model object for their study, as well as spermatozoa, the most famous flagellated cells that appeared along with sexual reproduction. Although cilia/flagella were first observed as early as in the 18th century, ciliary biology is still actively developing. New research perspectives that were until recently unthinkable are opening up with the development of high-resolution ultramicroscopy techniques, especially on living cells, as well as approaches that allow the structure of cilia/flagella to be assessed at the level of atomic resolution.
In the first part of the review, its authors note the fundamental principles of the structure and functions of cilia, and at the end they place special emphasis on modern achievements in ciliary biology using the latest imaging technologies.
Overall, I liked this review. It was written briefly, but at the same time capaciously, being easy and interesting to read. At the same time, I would like to address the authors with some comments/suggestions that, in my opinion, could improve the review, making it even more valuable and attractive to readers.
- Although the review mentions, for example, Chlamydomonasas a model system for studying cilia and flagella, it actually focuses mainly on biomedical applications, including human ciliary pathologies. In this regard, I think that the authors should either specify the title, limiting it to mammalian/human ciliated cells, or expand there view by adding a relevant section(s) on the diversity of flagella in eukaryotes and approaches to their study. The first is easier to do, the second would be more interesting. For example, the review does not discuss the evolutionary aspect at all. In some groups of organisms, it is the structure of sperm flagella that allows us to solve evolutionary-taxonomic questions, determine the mono- or polyphyletic origin of a group, and serves as a diagnostic criterion in taxonomy. If the title of the article is left as is, in a broad sense, then the authors' assertion that only motile sperm flagella have a 9 + 2 axoneme structure (e.g., lines 56–57; 179–180) is not entirely accurate. For example, not only the classic 9+2 pattern, but also the 9+0 structure and the specific 9+'1' trepaxonematan pattern characterize the sperm of flatworms. A bulk of work is devoted to this topic, but basically, as far as I know, they are carried out using conventional transmission electron microscopy (TEM). Another interesting example is the recent discovery of flagella in some amoeboid protists. Are modern imaging techniques used in this area and how can they help in solving issues of zoology, systematics and phylogeny?
Response to Reviewer: Thank you very much for your valuable comments. We fully agree with your recommendation to expand our manuscript by incorporating an evolutionary perspective. However, due to the limited timeframe provided for revisions as requested by the editor, we are constrained in our ability to adequately address the evolutionary aspects. Additionally, we recognize that our expertise does not specifically lie in cilia evolution, and we feel that any attempt to cover this topic comprehensively might not meet the rigorous standards expected. We are humbled to revise our title to “Advances in Imaging Techniques for Mammalian/Human Ciliated Cell’s Cilia: Insights into Structure, Function, and Dynamics”. We sincerely hope you find this adjustment acceptable and appropriate.
- Of course, the review should include a discussion of publications on the topic from the last yearor two. There are such works, including several reviews. For example, I think it would be worth mentioning the neuronal cilium, which would further attract the attention of readers with an emphasis on recent research.
Response to Reviewer: Thank you for your kind suggestions. We noticed that there indeed have several important findings of the role of cilia in neuronal cells and neurodegenerative disorders such as recently published papers. We added the related section as “3.4. Neuronal cilia and Nervous system diseases” (Line 288-310 in the revised manuscript). We sincerely hope you find these contents acceptable and appropriate.
- In my opinion, the review would have sparkled with completely different colors if its part devoted to the application of modern microscopy methods and achievements to study cilia/flagella had been supplemented with appropriate illustrations. I understand that authors may face permission issues, but the reader would have been able to immediately evaluate the advantages of this or that modern visualization method without referring to the original source.
Response to Reviewer: Thank you very much for your valuable comments and suggestions. As we previous mentioned in the response to reviewer 1, we find the timeframe (10 days) which the editor provided inadequate for securing the necessary permissions for the published images from both the original authors and the journals involved. We also drawn several new illustrations to improve the readability of our manuscript. The added illustrations are added in the section “4. Advancements in cilia imaging technology.” We believe that this illustration will help readers visualize the advantages of advanced imaging techniques.
- Regarding the methods used to study the fine structure of cilia, I would like to ask the authors whether, for example, immunoelectron microscopy (IEM), high-voltage electron microscopy, atomic force microscopy, including in vivo imaging, as well as correlative microscopy, which allows the structure of the same cell to be determined simultaneously at the light and electron level, are used for this purpose, and if so, to what extent?
Response to Reviewer: Thank you for your insightful comments regarding the methodologies for investigating the fine structure of cilia. Below, we tried to describe the potential applications of mentioned techniques to address your concerns:
Immunoelectron Microscopy (IEM): IEM is a useful technique that combines the advantages of immunolabeling and electron microscopy imaging to enable the localization of specific proteins to be studied at high resolution. It has been reported the applications for the protein localized on the cilia by using IEM. Wang et.al., used IEM to accurately localize the distribution of the TCTN1 protein at the base of the cilia when studying cilia transition zone proteins, revealing its Critical role in cilia assembly and signaling (Nat Commun 13, 3997 (2022)). Another study showed several proteins localized in the primary cilia of pancreatic islets by immunocolloidal gold technique combined with scanning electron microscopy, and also identified a variety of cilia-specific proteins, which provided an important basis for understanding the role of cilia in pancreatic islet function (J Cell Sci. 2024;137(20)).However, sample preparation procedures and the antibody specificity for IEM might be affecting the reliability of ultrastructural observations.
High-Voltage Electron Microscopy (HVEM): HVEM, a subtype of transmission electron microscopy (TEM), utilizes accelerating voltages typically between 500–1000 kV. This technique is advantageous due to its enhanced penetration depth, enabling the imaging of thicker specimens (up to 3–4 µm) and providing superior resolution for three-dimensional visualization of cellular and tissue structures. HVEM is particularly beneficial for studying interactions between cilia and their surrounding cellular environment, offering improved visualization of ciliary morphology and spatial localization. Nevertheless, HVEM necessitates specialized equipment, considerable technical expertise, and intensive data processing, limiting its accessibility and practical application in research settings (Phys. Rev. Lett.,2019:123, 150801). Despite its potential advantages, HVEM was not employed in our present study due to these practical constraints and because volume electron microscopy (vEM) sufficiently met our resolution and imaging requirements.
Atomic Force Microscopy (AFM): AFM is a high-resolution scanning probe technique widely utilized for analyzing surface topography and mechanical properties of biological samples. In cilia research, AFM might provide valuable insights into mechanotransduction properties by measuring force interactions or ionic current responses (Front. Bioeng. Biotechnol., 11 November 2021). Nonetheless, AFM has several limitations, including slow scanning speeds, susceptibility to thermal drift, and restricted magnification, which constrain its effectiveness for comprehensive ultrastructural analyses of cilia (Emerging Topics in Life Sciences (2021) 5 103–111). We acknowledge its potential utility when combined with complementary techniques in future investigations.
In Vivo Imaging and Correlative Microscopy: In vivo imaging techniques offer significant advantages in cilia research, providing dynamic observations of ciliary behavior, structure, and function under physiologically relevant conditions (Scientific Reports,2015;5:13216). Similarly, correlative microscopy (CLEM), which integrates fluorescence microscopy (FM) and electron microscopy (EM), is highly beneficial for simultaneously capturing functional dynamics and high-resolution structural details, thus enhancing our understanding of ciliopathies (Cilia: Methods and Protocols, Methods in Molecular Biology, 2016, vol. 1454; Nat Methods. 2015 Jun;12(6):503-13). Despite these advantages, both techniques involve substantial technical complexity, high operational costs, stringent experimental condition controls, and intricate data processing requirements. Additionally, CLEM faces specific challenges such as fluorescence quenching, complicated sample preparation protocols, and relatively slow imaging throughput (Methods in Cell Biology,Volume 162, 2021, Pages 1-11).
We sincerely appreciate your recommendations to consider these diverse imaging modalities and fully recognize their potential contributions toward achieving a more comprehensive understanding of ciliary structure and function. While these techniques indeed offer powerful and complementary capabilities, their specific applications and limitations require careful consideration aligned with research objectives.
- Important note. Section 4 needs to be restructured/rewritten.
The Section 4.4. “Optical imaging technology” should be moved into the Section 4.1 “Optical imaging”.
Similarly, aren’t cryo-EM (Sect. 4.3) or volume EM/ET (Sect 4.5) parts of the more extensive and diverse field of electron microscopy (Sect. 4.2)?
Therefore, I would suggest that the authors divide Section 4 into two large sections — Optical imaging and Electron microscopy imaging — with several relevant subsections within.
Finally, in addition to the above, lines 336–342 seem out of place in the Section “Electron microscopy”, since Ref. [94] (Rix et al., 2011) contains no EM data.
Response to Reviewer: Thank you for your valuable suggestions regarding the restructuring of Section 4 of our manuscript. Your insights have helped to improve the organization and clarity of our work. We have reorganized and revised section 4 accordingly (Line 311-538 in the revised manuscript).
We agree with your comment that the reference to Rix et al. 2011 in lines 336-342 was not appropriately placed in the “Electron Microscopy” section, and we deleted them to ensure that everything is consistent with the electron microscopy theme. We sincerely hope you find these contents acceptable and appropriate.
Additional minor points
- Lines 89–91: The cited work [21] is not a “recent” study, it appeared in 2014.
Response to Reviewer: Thank you for your kind comments. We replaced it with a recent published paper (2019) [Line 101-103, in revised manuscript].
- Line 120: The abbreviation PCM should be expanded upon when it is first mentioned here, and not only in Fig. 3 and in the caption to it.
Response to Reviewer: Thank you for your suggestions. The abbreviation PCM has been changed to the full name “pericentriolar material” in the first mention accordingly [Line 132, yellow highlighted].
- Lines 269–270: For a wide range of readers it would be useful to briefly explain the essence of the method OCT.
Response to Reviewer: Thank you very much for your valuable suggestions. We have provided a brief and clear explanation of the nature of the OCT method in the revised manuscript. In lines 328-331, we have added a description of the fundamentals of the OCT method to help readers better understand its application and importance in the study.
- Line 323: The name of the genus Parameciumshould be italicized.
Response to Reviewer: Revised accordingly.
- Please check your in-text citations, as it is customary to cite a work by its first author, not its last.If the author is the last one in the publication, you can write: “Someone’s research group”, but not “Someone et al.” This can be misleading.Below are some of the citation inconsistencies I have found: Line 392: Ref. 110 is Fang et al., not Yan et al.
Line 416: Yao Cong et al. is not in the Reference list.
Lines 417–422: Gaia et al. is not in the Reference list. However, the sentence (“This discovery...”) ends with the reference [117] to the work by Kiesel et al.
Line 426: Hughes is the last author of the work Polino et al. [119].
Response to Reviewer: Sorry for our mistakes. We corrected the author citations in the manuscript accordingly, and the exact location of the changes can be found in the revised manuscript.
- Electron microscopy does not deal with living cells (cf. line 401), only with fixed ones.
Response to Reviewer: Thank you for your comment and we sincerely apologize for our mistakes. We revised accordingly [Line 393, yellow highlighted].
Reviewer 3 Report
Comments and Suggestions for Authors
The manuscript entitled “Advances in Imaging Techniques for Cilia: Insights into Structure, Function, and Dynamics“ overviewed the recent technology for the cilia studies. They described the applications of super-resolution microscopy, electron microscopy, volume electron microscopy and cryo-EM/ET in cilia research from basic to clinical studies. Overall, they provided a comprehensive understanding of the cilia organization and functional mechanisms elucidated by advanced imaging techniques. The manuscript is well-written, but there are some issues that should be addressed.
Comments:
- In the “Research on cilia in organs and organ systems” parts, authors summarized some advances in sperm flagellum and ED. However, recent studies found the cilia are involved in the spermatogenesis and the meiotic process such as the published manuscripts (10.1126/science.abh310) and (10.1093/jmcb/mjac049) . Please adjust this section and integrate the recent updates.
- In line 81, “perturbationt” should be “perturbation”? please carefully recheck it.
- In line 106 and 130, the “Preciliary” should be “perciliary”.
- In line 152, what is the meaning of “situs inversus totalis”? please reconfirm the description.
Author Response
The manuscript entitled “Advances in Imaging Techniques for Cilia: Insights into Structure, Function, and Dynamics“ overviewed the recent technology for the cilia studies. They described the applications of super-resolution microscopy, electron microscopy, volume electron microscopy and cryo-EM/ET in cilia research from basic to clinical studies. Overall, they provided a comprehensive understanding of the cilia organization and functional mechanisms elucidated by advanced imaging techniques. The manuscript is well-written, but there are some issues that should be addressed.
Comments:
- In the “Research on cilia in organs and organ systems” parts, authors summarized some advances in sperm flagellum and ED. However, recent studies found the cilia are involved in the spermatogenesis and the meiotic process such as the published manuscripts (10.1126/science.abh310) and (10.1093/jmcb/mjac049) . Please adjust this section and integrate the recent updates.
Response to Reviewer: We sincerely appreciate the valuable comments. We have checked the literature carefully and added related contexts on the cilia involved in the spermatogenesis into the section 3.1. in our revised manuscript.
- In line 81, “perturbationt” should be “perturbation”? please carefully recheck it.
Response to Reviewer: We sincerely thank you for your careful reading. As suggested by the reviewer, we have corrected the“perturbationt" into“perturbation"in line 93.
- In line 106 and 130, the “Preciliary” should be “perciliary”.
Response to Reviewer: We sincerely thank you for your careful reading with reference to previous literature such as published manuscripts (Semin Cell Dev Biol. 2021;110:70-88) and (Nat Cell Biol, 2018; 20, 175–185) The phrase preciliary vesicles (PCVs) is a fixed phrase.
- In line 152, what is the meaning of “situs inversus totalis”? please reconfirm the description.
Response to Reviewer: We sincerely apologize for our mistakes. In our revised manuscript, we corrected “situs inversus totalis” to “situs inversus”. It is an uncommon reversal of organs in the body in which the apex of the heart points to the right and the liver and appendix are on the left side.
Round 2
Reviewer 1 Report
Comments and Suggestions for Authors
Unfortunately, the authors did not address my recomendation, which I repeat below.
However, I cannot recommend its publication in the current form, because it lacks illustrative material. There are only 4 schemes (Fig 1-4), however one would expect to have other images, showing the cilia and its functions. For example, the whole chapter 4. Advancements in cilia imaging technology does not feature a single microscopy image, which is definitely weird. The authors are requested to select appropriate previously published images, obtain the permission to republish and arrange them into figures. After that, the paper can be re-reviewed.
I assume that 10 days might be not enough for this update, but in this case the authors should have applied for an extension. In any case, in its current form the review article is not recommended for publication. I suggest the editor to reject it and encourage the authors to revise it according to my comment, and then, after it is ready, submit it again to this journal.
Reviewer 2 Report
Comments and Suggestions for Authors
I thank the authors for their careful consideration of my comments. I am completely satisfied with their accurate answers to the questions in my review and really appreciate the effort and work they put into improving the MS. New sections have been added to it and the structure has become more logical. The title has been changed and now better matches the content, purpose and objectives of the MS. In my opinion, the new figures added to illustrate the principles of advanced techniques in the field of cilia and flagella research/visualization seem very useful.
The only thing I would ask the authors to do is to correct a few more minor inaccuracies before the article is finally accepted for publication.
Minor points
- Lines 354–356: “Jiang et.al. have developed innovative fibronectin staining techniques for that offer improved resolution and reduced background noise, opening new avenues for studying cilia morphology[101].” Please check the citation of article [101] carefully. First, Pan T., not Jiang, is the first author of this article; second, this article mentions tubulin, not fibronectin.
- In section 4.2. “Electron Microscopy (EM) imaging”, subsection 4.2.1 “EM” I propose to rename to Transmission EM. This is necessary for clarification, as is done in the subsequent subsections of this section devoted to electron microscopy.
- In Fig. 6, please replace a cartoon of the microscope with an adequate one, since it is a light microscope rather than electron microscope in the picture.